EMBO
Molecular Medicine

# Beclin-1-mediated activation of autophagy improves proximal and distal urea cycle disorders

Leandro R Soria[1] [ID], Sonam Gurung[2], Giulia De Sabbata[3], Dany P Perocheau[2] [ID], Angela De Angelis[1], Gemma Bruno[1], Elena Polishchuk[1], Debora Paris[4], Paola Cuomo[4], Andrea Motta[4], Michael Orford[2], Youssef Khalil[2] [ID], Simon Eaton[2], Philippa B Mills[2], Simon N Waddington[2,5], Carmine Settembre[1] [ID], Andrés F Muro[3] [ID], Julien Baruteau[2,6] [ID] & Nicola Brunetti-Pierri[1,7,*] [ID]

## Abstract

Urea cycle disorders (UCD) are inherited defects in clearance of waste nitrogen with high morbidity and mortality. Novel and more effective therapies for UCD are needed. Studies in mice with constitutive activation of autophagy unravelled Beclin-1 as *drug-gable* candidate for therapy of hyperammonemia. Next, we investigated efficacy of cell-penetrating autophagy-inducing Tat-Beclin-1 (TB-1) peptide for therapy of the two most common UCD, namely ornithine transcarbamylase (OTC) and argininosuccinate lyase (ASL) deficiencies. TB-1 reduced urinary orotic acid and improved survival under protein-rich diet in *spf-ash* mice, a model of OTC deficiency (proximal UCD). In *Asl[Neo/Neo]* mice, a model of ASL deficiency (distal UCD), TB-1 increased ureagenesis, reduced argini-nosuccinate, and improved survival. Moreover, it alleviated hepatocellular injury and decreased both cytoplasmic and nuclear glycogen accumulation in *Asl[Neo/Neo]* mice. In conclusion, Beclin-1-dependent activation of autophagy improved biochemical and clinical phenotypes of proximal and distal defects of the urea cycle.

**Keywords** argininosuccinic aciduria; autophagy; OTC deficiency; Tat-Beclin-1 peptide; urea cycle disorders
**Subject Categories** Autophagy & Cell Death; Genetics, Gene Therapy & Genetic Disease

## Introduction

Autophagy is highly active in liver. Proteins, glycogen and lipid droplets are degraded by autophagy in liver cells to release amino acids, glucose and free fatty acids that can be reused for synthesis of new proteins and macromolecules, or can enter the tricarboxylic acid (TCA) cycle to generate ATP (Kaur & Debnath, 2015). Liver autophagy was recently found to support ammonia detoxification by furnishing the urea cycle with intermediates and energy that increase urea cycle flux under conditions of excessive ammonia (Soria *et al*, 2018). Liver-specific deficiency of autophagy impaired ammonia detoxification whereas its enhancement resulted in increased urea synthesis and protection against hyperammonemia (Soria *et al*, 2018). Therefore, drugs enhancing autophagy have potential for treatment of urea cycle disorders (UCD) (Soria & Brunetti-Pierri, 2018, 2019). In a previous study (Soria *et al*, 2018), we showed that rapamycin reduces orotic acid in *spf-ash* mice, a mouse model of the ornithine transcarbamylase (OTC) deficiency that carries a single nucleotide mutation in the fourth exon of the *Otc* gene resulting in a splicing defect and 10% of residual enzyme activity (Hodges & Rosenberg, 1989). Although it has been efficiently used to promote autophagy, rapamycin does not completely inhibit its target, the mechanistic target of rapamycin kinase complex 1 (mTORC1), and affects several biological processes besides autophagy (Li *et al*, 2014). Therefore, drugs targeting autophagy more specifically are attractive because they are expected to have less side effects. Tat-Beclin-1 (TB-1) is an engineered cell-permeable peptide that potently and specifically induces autophagy (Shoji-Kawata *et al*, 2013). TB-1 is formed by the HIV-1 Tat protein transduction domain attached via a diglycine linker to a peptide derived from Beclin-1 (*Becn1*), a key component of the autophagy induction machinery (Shoji-Kawata *et al*, 2013). In summary, TB-1 is an attractive therapeutic candidate for its specificity and at least in mice, it has shown great potential for treatment of various diseases, including several types of cancer, infections, cardiac dysfunction, skeletal disorders and axonal injuries (Cinque *et al*, 2015; He *et al*, 2016; Pietrocola *et al*, 2016; Bartolomeo *et al*, 2017;

1   Telethon Institute of Genetics and Medicine, Pozzuoli, Italy
2   UCL Great Ormond Street Institute of Child Health, London, UK
3   International Centre for Genetic Engineering and Biotechnology, Trieste, Italy
4   Institute of Biomolecular Chemistry, National Research Council, Pozzuoli, Italy
5   Wits/SAMRC Antiviral Gene Therapy Research Unit, Faculty of Health Sciences, University of the Witwatersrand, Johannesburg, South Africa
6   Metabolic Medicine Department, Great Ormond Street Hospital for Children NHS Foundation Trust, London, UK
7   Department of Translational Medicine, Federico II University, Naples, Italy
    *Corresponding author. Tel: +39 081 19230661; Fax: +39 081 5609877; E-mail: brunetti@tigem.it

Song *et al*, 2018; Sun *et al*, 2018; Vega-Rubin-de-Celis *et al*, 2018). In the present study, we investigated the therapeutic potential of TB-1 for treatment of UCD.

# Results

## Constitutional hyperactivation of Beclin-1 enhances ammonia detoxification

Beclin-1 is a central player in autophagy and regulates autophagosome formation and maturation (Liang *et al*, 2008). To investigate *Becn1* functions *in vivo*, a knock-in mouse model carrying a *Becn1* mutation (*Becn1^F121A^*) resulting in constitutively active autophagy has been recently generated (Rocchi *et al*, 2017). In these mice, Phe121 is mutated into alanine resulting in disruption of the BECN1-BCL2 binding and constitutive activation of BECN1 and autophagy in multiple tissues, including liver (Rocchi *et al*, 2017; Fernandez *et al*, 2018; Yamamoto *et al*, 2018). In these mice, we investigated ammonia detoxification by measurements of blood ammonia levels during acute hyperammonemia induced by an ammonia challenge. Despite no changes in blood ammonia at baseline, *Becn1^F121A^* mice showed 32% reduction in blood ammonia at 30 min after intraperitoneal (i.p.) injection of ammonium chloride compared to age-matched wild-type (WT) mice (Fig 1A). Accordingly, *Becn1^F121A^* mice showed enhanced ureagenesis compared to WT controls, as shown by increased blood levels of $^{15}$N-labelled urea from $^{15}$N-ammonium chloride (Fig EV1A). Improved ammonia clearance was not dependent on increased expression of urea cycle enzymes in *Becn1^F121A^* mice that showed similar enzyme levels by Western blotting compared to WT controls (Fig EV1B and C). Therefore, consistent with previous findings (Soria *et al*, 2018), gain-of-function mutation of the autophagy activator *Becn1* protects against acute hyperammonemia *in vivo*, suggesting that Beclin-1 is a *druggable* candidate for therapy of hyperammonemia.

## Tat-Beclin-1 improves the phenotype of OTC-deficient mice

To investigate the therapeutic efficacy of *Becn1*-mediated induction of autophagy in mouse models of UCD, we injected TB-1 i.p. in *spf-ash* mice (Hodges & Rosenberg, 1989), a model of OTC deficiency, the most common UCD. Body weights were unaffected by TB-1 (Fig EV2A). Although not normalized, in *spf-ash* mice the levels of the biochemical hallmark of OTC deficiency, urinary orotic acid, were significantly reduced by TB-1 (Fig 1B). Consistent with its autophagy enhancer activity, TB-1 increased the hepatic autophagic flux, as showed by reduced protein levels of the autophagosome marker LC3-II, and the two main autophagy cargo receptors, namely p62 and NBR1 (Fig 1C and D). Notably, OTC residual enzyme activity was unaffected by TB-1 (Fig EV2B), thus excluding reduction of urinary orotic acid as a consequence of increased residual OTC activity induced by TB-1. To further investigate the efficacy of TB-1-mediated increased liver autophagy for therapy of OTC deficiency, *spf-ash* mice were fed for 10 days with a high protein diet and were either treated with TB-1 or left untreated. Consistent with previous reports (Yang *et al*, 2016; Kurtz *et al*, 2019), *spf-ash* mice showed marked mortality under high protein diet compared to control WT mice (Fig 1E). An approximately 30% weight loss was observed in

all *spf-ash* mice fed with the high protein diet, independently of TB-1 treatment (Fig EV2C). Although it did not significantly improved survival as single treatment, when combined with an ammonia scavenger drug (Na-benzoate) and L-arginine (L-Arg), TB-1 increased survival whereas ammonia scavengers and L-Arg did not affect survival of *spf-ash* mice under high protein diet (Fig 1E). Consistent with the increased survival, blood ammonia levels measured after 4 days with high protein diet (a time-point prior to mortality) were significantly lower in *spf-ash* mice treated with the combination of TB-1 and Na-benzoate and L-Arg compared to untreated *spf-ash* mice (Fig 1F). TB-1 alone decreased slightly but not significantly blood ammonia whereas Na-benzoate and L-Arg significantly reduced blood ammonia levels, consistent with the human data (Enns *et al*, 2007; Fig 1F). Notably, Na-benzoate and L-Arg treatment did not affect the levels of urinary orotic acid increased by the high protein diet whereas TB-1 either alone or in combination with Na-benzoate and L-Arg efficiently blunted the increase in urinary orotic acid induced by the high protein diet (Fig EV2D). Taken together, these results support the therapeutic potential of activation of liver autophagy by TB-1 in combination with conventional treatments, such as ammonia scavenger drugs and L-Arg (Enns *et al*, 2007; Haberle *et al*, 2019), for treatment of OTC deficiency, the most common UCD.

## Tat-Beclin-1 enhances ureagenesis and corrects metabolic abnormalities of argininosuccinic aciduria

To investigate the efficacy of autophagy enhancement for therapy of argininosuccinic aciduria (ASA), the second most frequent UCD (Baruteau *et al*, 2019a), we investigated TB-1 treatment in the hypomorphic murine model of argininosuccinate lyase (ASL) deficiency (*Asl^Neo/Neo^*) that expresses approximately 16% of residual enzyme activity and recapitulates the main biochemical and clinical abnormalities of ASA patients (Erez *et al*, 2011; Nagamani *et al*, 2012; Baruteau *et al*, 2018; Burrage *et al*, 2020). Besides impaired urea synthesis and ammonia detoxification, systemic manifestations of ASA, such as reduced body weight, increased blood pressure, and reduced survival are also associated with nitric oxide (NO)-deficiency (Erez *et al*, 2011; Nagamani *et al*, 2012; Baruteau *et al*, 2018; Kho *et al*, 2018). *Asl^Neo/Neo^* mice treated with TB-1 but without any additional treatment showed increased survival compared to vehicle-treated controls that started dying by 10 days of age (Fig 2A). Weight gain was unaffected by TB-1 (Fig EV3A). Consistent with our previous work (Soria *et al*, 2018), TB-1-mediated activation of autophagy in *Asl^Neo/Neo^* mice was associated with increased incorporation of $^{15}$N into urea (+88%, $P < 0.05$) indicating enhanced ureagenesis (Fig 2B). Consistent with the increased ureagenesis, blood ammonia levels were lowered by TB-1 in *Asl^Neo/Neo^* mice (Fig EV3B). As expected, autophagic flux was enhanced in livers of *Asl^Neo/Neo^* mice injected with TB-1, as shown by reduced LC3-II and decreased autophagy substrates (p62 and NBR1) in livers (Fig 2C and D), whereas residual ASL enzyme activity was unaffected (Fig EV3C). Argininosuccinic acid levels were reduced in dried blood spots (Fig 2E) in TB-1-treated *Asl^Neo/Neo^* mice. Consistent with this reduction, hepatic content of $^{15}$N-labelled argininosuccinic acid was also reduced in mice treated with TB-1 (Fig 2F). Metabolomic analysis by $^1$H-NMR spectroscopy (Soria *et al*, 2018) showed that the whole-liver metabolome of vehicle-treated *Asl^Neo/Neo^* mice was

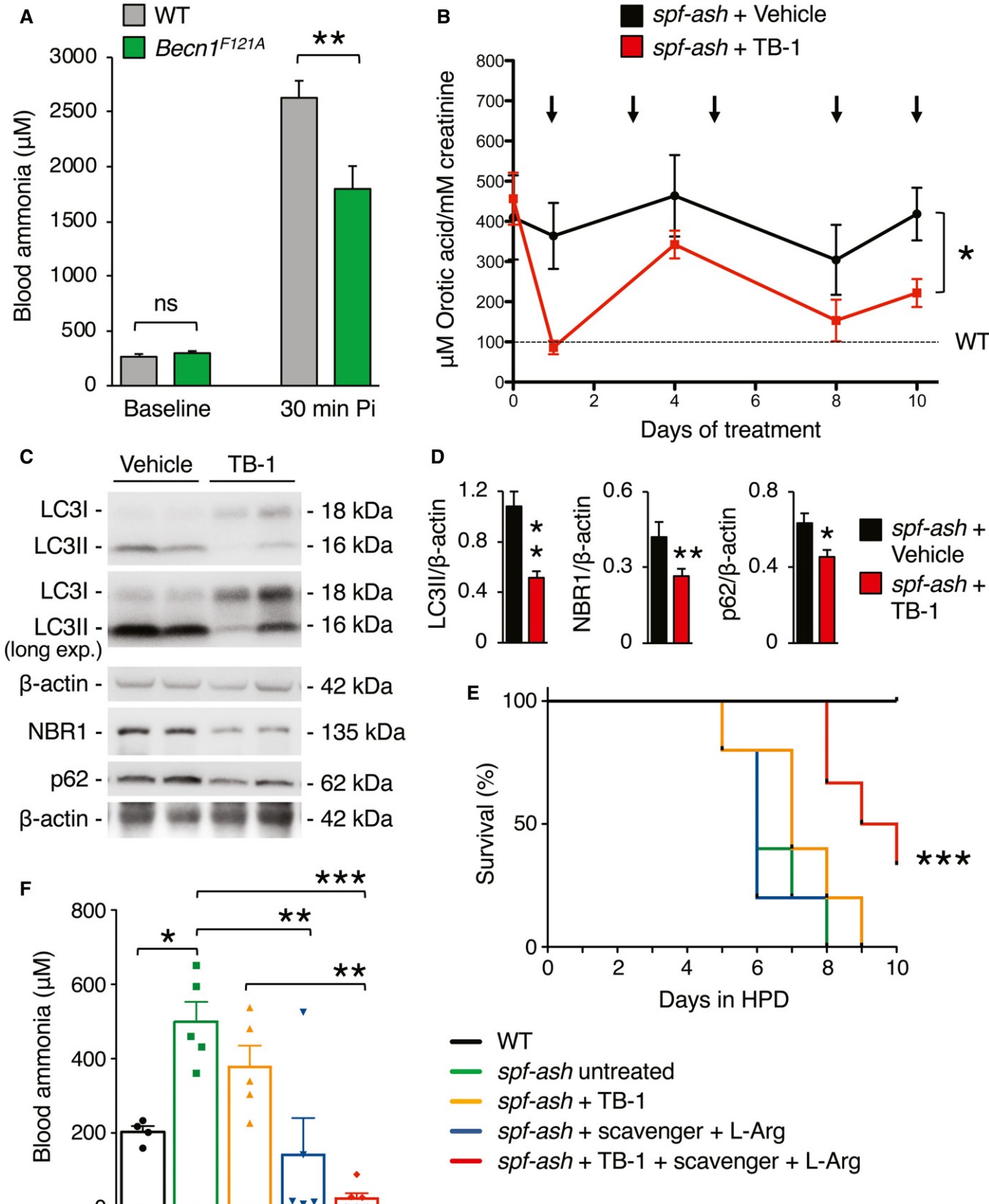

**Figure 1.**

◀

**Figure 1.  Hyperactive Beclin-1 protects against hyperammonemia, and activation of hepatic autophagy improves the phenotype of OTC-deficient mice.**

A    Blood ammonia in 8–9-week-old C57BL/6J wild-type (WT) mice ($n = 8$) and $Becn1^{F121A}$ mice with constitutive activation of autophagy ($n = 8$) at baseline and 30 min after i.p. injection of $NH_4Cl$ (10 mmol/kg). $**P < 0.01$ (Unpaired $t$-test). ns: not statistically significant difference.

B    Urinary orotic acid of 12-week-old *spf-ash* mice treated with TB-1 (15 mg/kg, i.p) or vehicle at various times as indicated by the arrows ($n = 5$ mice/group). $*P < 0.05$ (Two-way ANOVA).

C, D  Western blotting and densitometric quantifications of autophagy markers (LC3II: autophagosomes; p62 and NBR1: cargo receptors) in livers of *spf-ash* mice harvested after 10 days of treatment with TB-1 or vehicle. β-actin was used as loading control. $n = 5$ mice/group. $**P < 0.01$, $*P < 0.05$ (Unpaired $t$-test).

E    Survival curves of *spf-ash* mice fed with a high protein diet (HPD) for 10 days and treated with TB-1 alone or combined with scavenger drug (Na-benzoate) and L-Arginine, or treated with scavenger drug (Na-benzoate) and L-Arginine, or left untreated. WT control were included ($n = 5$/group). $***P < 0.001$ (Log-rank Mantel–Cox test).

F    Blood ammonia levels determined after 4 days under HPD ($n = 5$ mice/group); $***P < 0.001$, $**P < 0.01$, $*P < 0.05$ (One-way ANOVA).

Data information: Treatments in (E, F): Scavenger (Na-benzoate 250 mg/kg/day, i.p.) and L-arginine (L-Arg, 250 mg/kg/day, i.p.); TB-1 (15 mg/kg every 2 days, i.p). WT mice were age-, gender- and strain- (C3H)-matched. All values are shown as averages ± SEM. Exact $P$ values are reported in Appendix Table S1.
Source data are available online for this figure.

well separated from healthy WT controls but it was shifted towards non-diseased WT controls in $Asl^{Neo/Neo}$ mice injected with TB-1 (Fig 2G and Appendix Fig S1), suggesting that TB-1 corrects at least partially the liver metabolic deregulation caused by ASL deficiency. Notably, NMR confirmed that liver content of argininosuccinate, along with its two precursors citrulline and aspartate, was reduced (Fig EV4). Moreover, levels of key compounds of the TCA cycle (fumarate and succinate) and glucose were rescued by TB-1 (Fig EV4). In summary, TB-1 improved several biochemical alterations of ASA, confirming the efficacy of autophagy enhancer molecules for therapy of UCD.

**Tat-Beclin-1 reduces injury and abnormal glycogen deposition in livers with ASL deficiency**

Chronic hepatocellular injury is a common complication in patients with ASL deficiency (Mori *et al*, 2002; Yaplito-Lee *et al*, 2013; Baruteau *et al*, 2017; Ranucci *et al*, 2019). Despite the underlying mechanism triggering the liver disease remains unclear, evidence in human and mouse suggests that it is related to massive accumulation of cytoplasmic glycogen (Badizadegan & Perez-Atayde, 1997; Bigot *et al*, 2017; Burrage *et al*, 2020). Moreover, because activation of autophagy was found to be effective in clearance of glycogen storage in glycogen storage diseases (Ashe *et al*, 2010; Spampanato *et al*, 2013; Martina *et al*, 2014; Farah *et al*, 2016), we investigated whether TB-1 promotes glycogen clearance in ASA livers. To this end, $Asl^{Neo/Neo}$ mice received protein-restricted diet and daily administration of Na-benzoate and L-Arg in combination with either TB-1 or vehicle, started on day 10 of life and lasting for 3 weeks. Consistent with previous data (Erez *et al*, 2011; Ashley *et al*, 2018; Baruteau *et al*, 2018; Burrage *et al*, 2020), vehicle-treated $Asl^{Neo/Neo}$ mice showed vacuolated hepatocyte cytoplasm by haematoxylin and eosin (H&E) staining in contrast to WT mice, whereas TB-1 treatment markedly improved the microscopic changes of liver architecture (Fig 3A). Although body weight was unaffected (Fig EV5A), TB-1 treatment resulted in a trend of reduction in hepatomegaly (Fig EV5B) and a mild decrease in serum alanine aminotransferase (ALT) levels in $Asl^{Neo/Neo}$ mice (Fig EV5C). Moreover, $Asl^{Neo/Neo}$ mice treated with TB-1 showed partial reduction of liver glycogen storage by periodic acid Schiff (PAS) staining (Fig 3A and B) and glycogen quantification (Fig 3C) compared to controls. Notably, glycogen accumulation was not observed in livers of *spf-ash* mice (Fig EV5D). Glycogen in hepatocytes is catabolized either in cytosol

by the coordinated action of enzymes involved in glycogenolysis or in the lysosome by the acid glucosidase (Prats *et al*, 2018). Hepatic expression of glycogen phosphorylase (PYGL), the enzyme that catalyses the rate limiting step of glycogenolysis, was recently found to be reduced in $Asl^{Neo/Neo}$ mice, suggesting a mechanism responsible for aberrant glycogen accumulation in ASA (Burrage *et al*, 2020). We confirmed reduced PYGL protein levels in livers of $Asl^{Neo/Neo}$ mice, but they were unaffected by TB-1 (Fig 3D), suggesting that reduction of glycogen by TB-1 does not occur through rescue of the cytosolic glycogen degradation pathway. Moreover, when autophagy flux is increased by TB-1, glycogen degradation in lysosomes is efficiently achieved, as confirmed by electron microscopy (EM) analysis showing clearance of cytoplasmic glycogen accumulation (Fig 3E). Moreover, glycogen accumulation in $Asl^{Neo/Neo}$ mice resulted in displacement of organelles to the cell membrane as previously reported (Burrage *et al*, 2020), that is relieved by TB-1 (Fig 3E). Surprisingly, EM analysis also showed abundant intranuclear glycogen deposition in hepatocytes of $Asl^{Neo/Neo}$ mice, that was partially reduced by TB-1 (Fig 4A and B). In summary, in addition to promoting urea synthesis, TB-1 reduced abnormal glycogen storage in cytosol and nuclei of ASL-deficient hepatocytes.

# Discussion

UCD are inborn errors of metabolism due to impaired clearance of toxic nitrogen. Despite current therapies, cumulative morbidity is still high in patients with UCD and thus, several experimental therapies have been investigated to improve clinical outcomes (Soria *et al*, 2019). We recently showed a role of hepatic autophagy in promoting ureagenesis and ammonia detoxification (Soria *et al*, 2018) that can be exploited for the development of novel therapies for hyperammonemia and UCD (Soria *et al*, 2018; Soria & Brunetti-Pierri, 2018, 2019). In the present study, we investigated the efficacy of TB-1 peptide, a potent and specific agent that can activate autophagy *in vivo*, in two well-established mouse models of proximal and distal UCD. Induction of liver autophagy was found to improve several clinically relevant endpoints in these mice, supporting autophagy enhancement as a therapeutic strategy for UCD.

Autophagy plays a key role in liver physiology by supporting metabolism and promoting adaptation to stress. Specific modulation of autophagy has been recognized as a potential therapeutic strategy in various liver diseases (Allaire *et al*, 2019; Hazari *et al*, 2020).

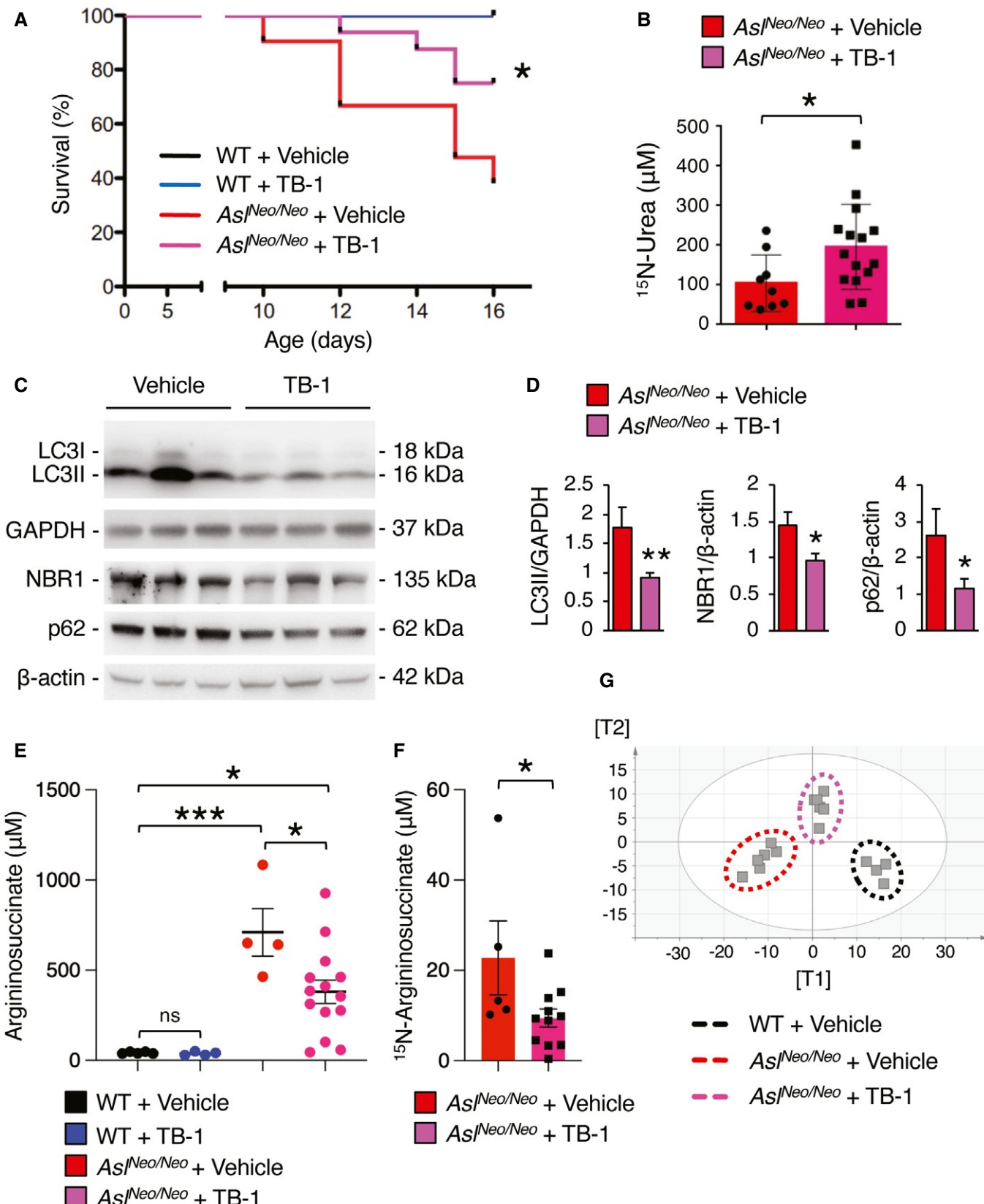

**Figure 2.**

**Figure 2. Enhancement of liver autophagy improves survival, increases ureagenesis, and corrects metabolic defects of ASL-deficient mice.**

A  Survival curves of $Asl^{Neo/Neo}$ mice and age-matched wild-type (WT) controls treated with TB-1 (15 mg/kg, i.p., every 48 h starting at day 10 of age) or vehicle. WT + Vehicle $n = 8$; WT + TB-1 $n = 8$; $Asl^{Neo/Neo}$ + Vehicle $n = 20$; $Asl^{Neo/Neo}$ + TB-1 $n = 16$. *$P < 0.05$ (Log-rank Mantel–Cox test).

B  Isotopic enrichment of $^{15}$N-labelled urea in blood, 20 min after i.p. injection of $^{15}NH_4Cl$ tracer (4 mmol/kg) in $Asl^{Neo/Neo}$ mice treated with TB-1 ($n = 15$) or vehicle ($n = 9$). *$P < 0.05$ (Unpaired $t$-test).

C  Representative Western blotting bands of LC3, p62 and NBR1 in livers of $Asl^{Neo/Neo}$ mice treated with TB-1 or vehicle. GAPDH and β-actin were used as loading controls.

D  Densitometric quantifications. $Asl^{Neo/Neo}$ + Vehicle $n = 5$; $Asl^{Neo/Neo}$ + TB-1 $n = 9$. **$P < 0.01$, *$P < 0.05$ (Unpaired $t$-test).

E  Argininosuccinate in dried blood spots of WT and $Asl^{Neo/Neo}$ mice injected with TB-1 or vehicle ($n = 4$–14 mice/group). ***$P < 0.001$, *$P < 0.05$ (One-way ANOVA).

F  Isotopic enrichment of $^{15}$N-labelled argininosuccinate in livers of $Asl^{Neo/Neo}$ mice treated with TB-1 ($n = 11$) or vehicle ($n = 5$). *$P < 0.05$ (Unpaired $t$-test).

G  Orthogonal Projection to Latent Structure-Discriminant Analysis (OPLS-DA) score plot obtained from high-resolution $^1$H-NMR spectroscopy performed on livers of vehicle-treated $Asl^{Neo/Neo}$ mice ($n = 6$), WT controls ($n = 4$) and $Asl^{Neo/Neo}$ mice injected with TB-1 ($n = 6$). A statistical model with $R^2 = 0.78$ (goodness of fit), $Q^2 = 0.57$ (power in prediction) and $P = 0.0056$ was obtained. See also Appendix Fig S1.

Data information: All values are shown as averages ± SEM. ns: not statistically significant difference. Exact $P$ values are reported in Appendix Table S1.
Source data are available online for this figure.

Obtaining specific modulation of autophagy has been a major challenge for clinical translation of autophagy enhancer molecules (Allaire *et al*, 2019). Being an essential regulator of autophagosome synthesis and maturation, Beclin-1 is an attractive target for autophagy-inducing drugs and small peptides affecting Beclin-1 interactions, such as TB-1 and its derivatives have been developed (Shoji-Kawata *et al*, 2013; Peraro *et al*, 2017). Moreover, drugs affecting post-translational modifications of Beclin-1 altering its function are also attractive (Hill *et al*, 2019). The $Becn1^{F121A}$ mouse model used in this work mimics the condition of disrupted interaction of Beclin-1 with BCL2. In these mice, we found improved ureagenesis and increased ammonia detoxification capacity. Within the context of a normal functioning ureagenesis, the increase detected in $Becn1^{F121A}$ mice was relatively mild and corresponded to about 20–30%. Nevertheless, such increase is still expected to provide significant clinical benefit in UCD. Additionally, the increased ureagenesis of $Becn1^{F121A}$ mice validates the inhibitors of the Beclin-1-BCL2 complex (Chiang *et al*, 2018) as targets for increasing ammonia detoxification.

We next investigated the efficacy of TB-1-mediated activation of hepatic autophagy for therapy of proximal and distal UCD using the most relevant mouse models for these disorders, the *spf-ash* and the $Asl^{Neo/Neo}$ mice for OTC and ASL deficiency, respectively (Moscioni *et al*, 2006; Prieve *et al*, 2018; Kurtz *et al*, 2019; Soria *et al*, 2019). In *spf-ash* mice treated with TB-1, we detected reduced urinary orotic aciduria under standard and high protein diet. Importantly, combined with clinically available drugs enhancing ammonia excretion (Na-Benzoate) and urea synthesis (L-Arg), TB-1 was also effective in increasing survival of *spf-ash* mice challenged with a high

protein diet. These data suggest that liver autophagy enhancement cooperates with current treatment in improving the phenotype of OTC deficiency. In ASA mice, we also found that hepatic enhancement of autophagy increased ureagenesis and survival, and reduced argininosuccinate levels along with generalized rescue of the metabolic derangement, as suggested by liver metabolomic analyses. In summary, TB-1 improved the phenotypes of two UCD animal models, supporting the potential of autophagy enhancement for therapy of hyperammonemia due to defects of the urea cycle.

Increased liver autophagy by TB-1 provides amelioration but not a cure for UCD. The therapeutic potential of TB-1 could be especially exploited in newborns or infants with UCD in their early disease stages before more definitive treatments, such as liver transplantation or gene therapy can be safely and effectively performed. For liver-directed gene therapy with adeno-associated viral (AAV) vectors, efficacy is gradually lost during mouse growth as a consequence of dilution of episomal vector genomes in dividing hepatocytes (Cunningham *et al*, 2009; Baruteau *et al*, 2018). Hence, TB-1 can be viewed as a bridge treatment until affected infants reach an age that permits sustained transgene expression by gene therapy. Interestingly, co-administration of autophagy enhancers with AAV vectors markedly improved transgene expression (Hosel *et al*, 2017) and thus, TB-1 could have the dual action of increasing AAV transduction and improving metabolic control. Moreover, onset of hepatic gene expression by AAV typically requires 2–3 weeks, and thus, TB-1 might also treat the metabolic defect until high levels of gene expression are achieved.

Interestingly, in ASA mice TB-1 also resulted in increased clearance of intracellular glycogen accumulation. Disposal of

**Figure 3. Enhancement of autophagy reduces hepatocellular injury and glycogen storage in ASL-deficient mice.**

A  Haematoxylin and eosin (H&E, upper panels) and periodic acid Schiff (PAS, lower panels) staining of liver samples harvested from wild-type (WT) and $Asl^{Neo/Neo}$ mice treated with TB-1 or vehicle. Scale bars: 500 μm.

B  Computational analysis of PAS staining ($n \geq 4$ mice/group). **$P < 0.01$, *$P < 0.05$ (One-way ANOVA).

C  Quantification of hepatic glycogen in vehicle ($n = 6$)- and TB-1-treated $Asl^{Neo/Neo}$ mice ($n = 12$) compared to WT ($n = 5$) controls. **$P < 0.01$, *$P < 0.05$ (One-way ANOVA).

D  Representative Western blotting bands and densitometric quantification of PYGL in livers of WT and $Asl^{Neo/Neo}$ mice treated with either with TB-1 or vehicle ($n = 4$ mice/group). *$P < 0.05$ (One-way ANOVA). GAPDH was used as loading control.

E  Representative electron microscopy images of liver samples harvested from WT and $Asl^{Neo/Neo}$ mice treated with TB-1 or vehicle. Scale bar: 900 nm.

Data information: All values are shown as averages ± SEM. Exact $P$ values are reported in Appendix Table S1.
Source data are available online for this figure.

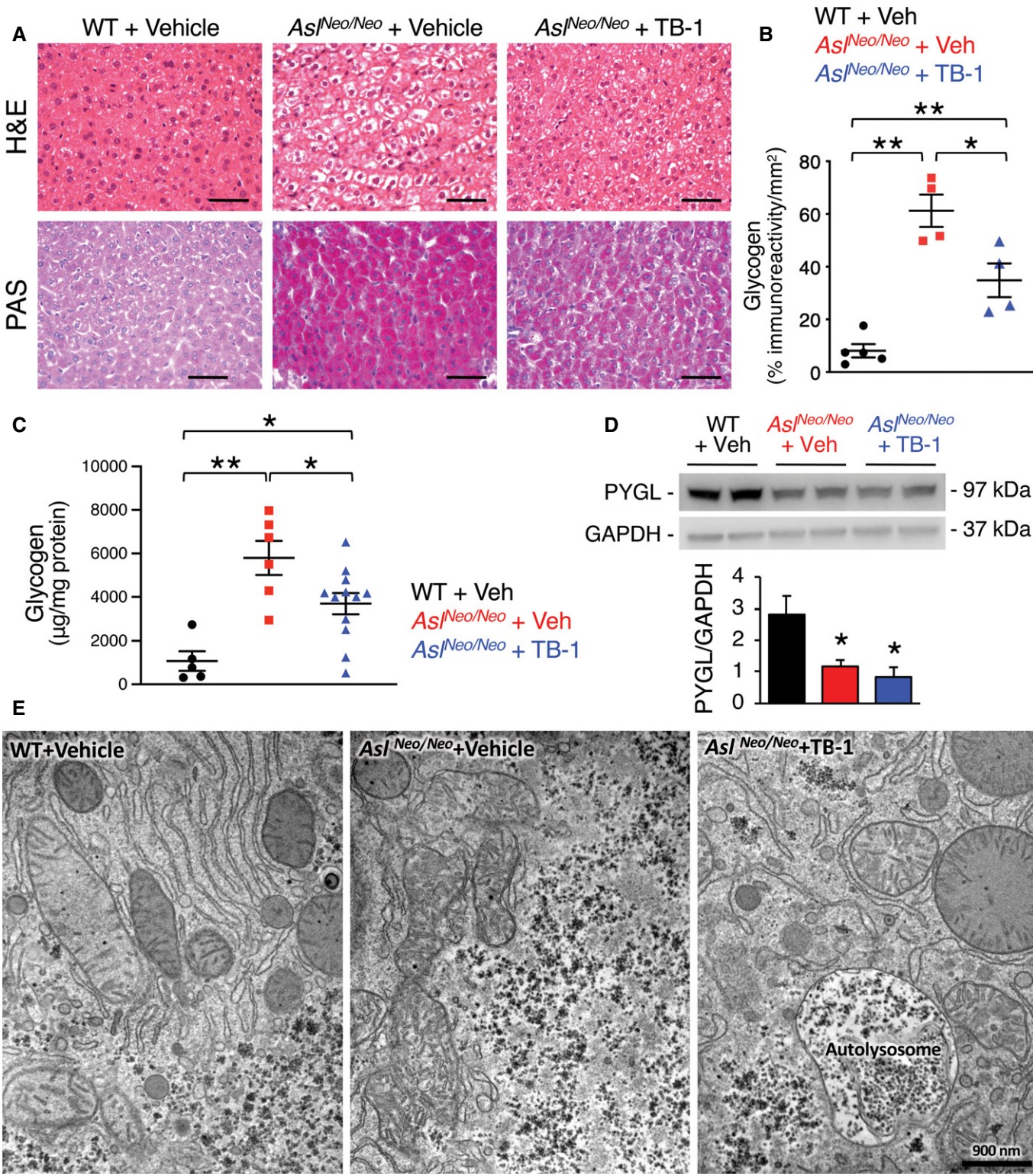

**Figure 3.**

carbohydrates such as glycogen via autophagy plays a crucial role in glucose homeostasis (Karsli-Uzunbas *et al*, 2014). This finding is consistent with previous studies in glycogen storage diseases type 1 and 2 showing that either genetic or pharmacologic activation of

autophagy reduced glycogen storage (Ashe *et al*, 2010; Spampanato *et al*, 2013; Martina *et al*, 2014; Farah *et al*, 2016). Consistent with a previous study (Burrage *et al*, 2020), we found that PYGL was reduced in ASL-deficient mice and thus, increased glycogen

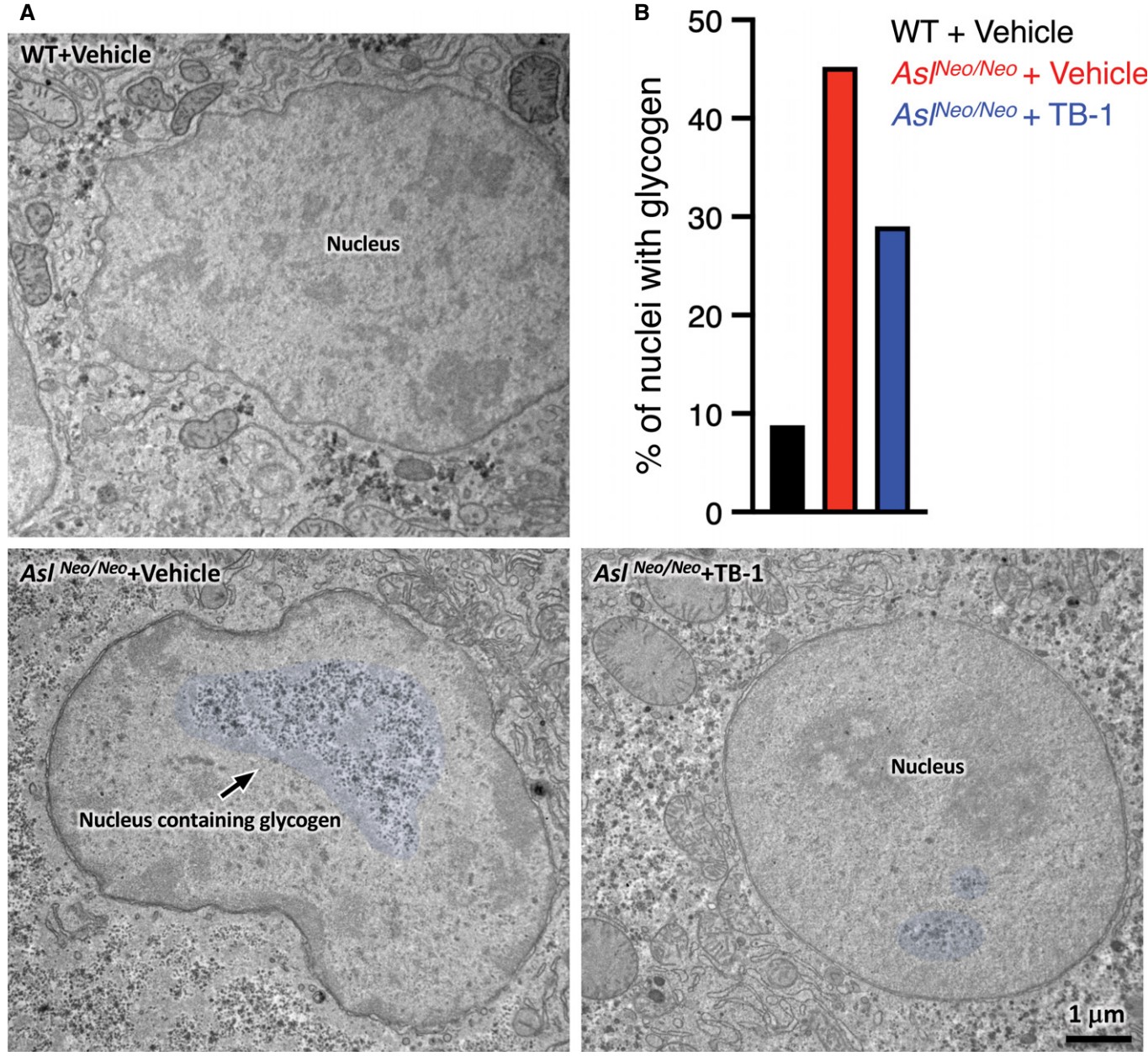

**Figure 4. TB-1 reduces intra-nuclear glycogen deposition in ASL-deficient mice.**

A  Representative electron microscopy images of liver samples harvested from wild-type (WT) and *Asl^{Neo/Neo}* mice treated with TB-1 or vehicle. False colour on the images indicates glycogen within the nuclei. Scale bar: 1 μm.

B  Quantification of nuclei containing glycogen (approx. 180 nuclei were analysed in total per condition, *n* = 3 mice/group).

Data information: All values are shown as averages ± SEM. ns: not statistically significant difference.

Source data are available online for this figure.

degradation mediated by TB-1 occurred independently from impaired cytosolic PYGL-dependent degradation of glycogen.

Glycogen clearance induced by TB-1 restored physiologic organelle distribution within the cell. Abnormal glycogen deposition in UCD livers occurs independently from metabolic syndrome (Bigot *et al*, 2017) and has been reported in patients with OTC deficiency and other UCD (Badizadegan & Perez-Atayde, 1997; Mori *et al*, 2002; Yaplito-Lee *et al*, 2013; Baruteau *et al*, 2017; Bigot *et al*, 2017;

Ranucci *et al*, 2019). However, *spf-ash* mice showed normal hepatic glycogen storage. Intriguingly, we found glycogen accumulation also in nuclei of ASA hepatocytes. Nuclear accumulation of glycogen was recently associated to epigenetic changes in gene expression and cancer (Sun *et al*, 2019). In normal cells, nuclear glycogenolysis provides a carbon pool for proper histone acetylation, whereas in cancer cells suppression of nuclear glycogen catabolism is associated with changes in histone acetylation and gene expression, and

cancer progression (Sun *et al*, 2019). Liver cancer is an emerging feature of UCD (Wilson *et al*, 2012; Koo *et al*, 2017; Wang *et al*, 2019). Therefore, besides improved ureagenesis, the reduced cytoplasmic and intra-nuclear glycogen mediated by TB-1 might provide further long-term clinical benefits preventing progression of chronic liver damage. However, this hypothesis would need further investigation and available mouse models of UCD have not been reported to have an increased frequency of liver cancer. The newly generated OTC knock-out mouse model that develops liver fibrosis and chronic liver damage (Wang *et al*, 2017) might be a suitable model to investigate this issue in long-term studies.

We previously found that TB-1 increases ureagenesis by furnishing the urea cycle with key intermediate metabolites and preventing ammonia-induced depletion of ATP (Soria *et al*, 2018). Whether this mechanism is also responsible for increased ureagenesis in ASL deficiency cannot be established. *Asl*[Neo/Neo] mice have massive accumulation of argininosuccinate secondary to deficiency of ASL that breaks down argininosuccinate into arginine and fumarate. In *Asl*[Neo/Neo] mice, TB-1 increased ureagenesis and reduced argininosuccinate despite the lack of increased ASL activity. Although no changes in ASL enzyme activity were detected, the enzyme assay might not reflect the enzyme *in vivo* activity. Alternatively, other mechanisms can be hypothesized. For instance, TB-1 might improve metabolic channelling, i.e. reversible specific assembly of enzyme clusters that accelerate processing of metabolite intermediates (Castellana *et al*, 2014; Pareek *et al*, 2020). Notably, channelling of urea cycle intermediates has been found between mitochondria and cytosol where ureagenesis takes place (Cheung *et al*, 1989; Cohen *et al*, 1992). This channelling could be particularly affected in *Asl*[Neo/Neo] mouse hepatocytes displaying aberrant distribution of their sub-cellular compartments that is partially corrected by TB-1, as confirmed by the ultrastructural studies performed in this work.

In conclusion, this study shows a key role of liver autophagy in nitrogen homeostasis and indicates that Beclin-1 as a *druggable* target for therapy of hyperammonemia and UCD. Moreover, these findings show that enhancement of hepatic autophagy is beneficial in UCD through: (i) correction of the underlying metabolic abnormalities by supporting residual ureagenesis activity and, (ii) reduction of the aberrant intracellular (cytosolic and nuclear) glycogen burden that might prevent long-term hepatotoxicity.

# Materials and Methods

### Mouse studies

All mouse procedures were performed in accordance with regulations and were authorized by either the Italian Ministry of Health or the UK Home Office. *Becn1*[F121A] (B6.129(Cg)-Becn1tm1.1Hec/J) mice were previously described (Rocchi *et al*, 2017; Fernandez *et al*, 2018) and were maintained on a C57BL/6 background. Wild-type (WT) littermates were used as controls. For acute ammonia challenges, male and female WT and *Becn1*[F121A] mice were starved overnight before the i.p. injection of 10 mmol/kg of [15]N-labelled ammonium chloride (98% enriched in [15]N, Sigma) dissolved in water. Blood samples were collected by retro-orbital bleedings at baseline, 5, 15, and 30 min post-injection. The amount of [15]N-labelled urea in sera was quantified by gas chromatography–mass

spectrometry (GC-MS) analysis at the Metabolic Core of The Children's Hospital of Philadelphia (Philadelphia, PA). Mice were sacrificed by cervical dislocation, and liver samples were harvested for analyses. Breeding pairs of *spf-ash* mice (B6EiC3Sn a/A-OTC[Spf-Ash]/J) were purchased from Jackson Laboratories (Cat# 001811) and housed in individually ventilated cages, maintaining a temperature of 22°C (± 2°C), relative humidity of 55% (± 10%), 15–20 air exchanges per hour and 12-h light/12-h dark cycle and receiving a standard chow diet and water *ad libitum*. 12-week-old male *spf-ash* mice received an i.p. injection of TB-1 D-11 retroinverso form peptide (Novus Biologicals; Cat# NBP2-49888) at the dose of 15 mg/kg [dissolved in phosphate buffer saline (PBS)] every 48 h for a total period of 10 days. Animals were daily monitored and weighted. Urine samples were collected at the indicated time points to measure orotic acid. A control group of *spf-ash* animals were injected with vehicle only. For high protein diet challenge, 12-week-old male *spf-ash* mice were maintained on a 51%-protein diet (U8959 version 142, Safe-diets) for 10 days. TB-1 peptide was delivered by i.p. injection at the dose of 15 mg/kg every 48 h. 60 mg/ml Na-Benzoate (Sigma; Cat# 18106) and 60 mg/ml L-Arg (Sigma, Cat# A5006) dissolved in PBS were i.p. injected at the dose of 250 mg/kg Na-Benzoate, 250 mg/kg L-Arg every 24 h. As controls, a group of *spf-ash* mice were injected with vehicle only and a group of WT mice were used. Animals were daily monitored and weighted. Blood samples were collected by submandibular bleeding at day 4 to measure ammonia concentration. Urines were collected at baseline and day 4 for measurements of orotic acid.

*Asl*[Neo/Neo] mice (B6.129S7-*Asl*[tm1Brle]/J) were purchased from Jackson Laboratory (Bar Harbor, ME) and maintained on standard rodent chow (Harlan 2018, Teklab Diets, Madison, WI; protein content 18%) with free access to water in a 12 h light/12 h dark environment. For mouse genotyping, DNA extraction from tail clips was performed as described previously (Baruteau *et al*, 2018). For long-term experiment, all WT and *Asl*[Neo/Neo] mice received a supportive treatment modified (Baruteau *et al*, 2018) including a reduced-protein diet (5CR4, Labdiet, St Louis, MO; protein content 14.1%) and daily i.p. injections of Na-benzoate (0.5 g/kg/day) and L-Arg (0.5 g/kg/day) from day 10 to day 30. Treated *Asl*[Neo/Neo] mice received 15 mg/kg of TB-1 i.p. three times per week (Monday, Wednesday, Friday). Control animals received vehicle (i.e. PBS). Male WT and *Asl*[Neo/Neo] mice received three i.p. injections over 5 or 20 days with either PBS or TB-1 from 10 days old onwards. Concomitantly, these mice received daily supportive treatment to improve survival until day 30. WT littermates were used as controls. For all experiments, WT and *Asl*[Neo/Neo] littermates were housed in the same cages. WT and mutant pups remained in cages with their mothers until day 30. Pups and their mothers were fed with the same diet. Mice had free access to diet and were not fasted before sacrifice.

### Biochemical measurements

Blood ammonia levels were measured by ammonia colorimetric assay kits (BioVision Incorporated; Cat# K370-100 or Sigma; Cat#AA0100) according to the manufacturer's instructions. Circulating ALT levels were analysed with a Fuji Dri-Chem NX500 (FUJIFILM, Tokyo, Japan). Orotic acid was purchased from Sigma-Aldrich. The isotopically labelled internal standard orotic acid was purchased from Cambridge Isotope Laboratories, Inc. The

quantitative experiments were done using as internal standard the isotopically labelled 1,3-$^{15}N_2$ orotic acid in 200μM concentration both for calibration curve and samples. A typical calibration curve ranged from 15 to 300 μM with excellent linearity ($R^2 > 0.99$). A Bruker (Bremen, Germany) amaZon SL bench-top ion trap mass spectrometer, equipped with an electrospray source, was employed for this study. The source was operated in negative ion mode with a needle potential of 4,500 V and a gas flow of 12 l/min of nitrogen with heating at 200°C. The chromatographic separations for quantitative experiments were performed using a series 1260 Agilent Technologies (Waldbronn, Germany) HPLC with auto sampler controlled from the Bruker Hystardata system. A Phenomenex (Torrance, USA) HPLC column Gemini C18 5 μm, 110 Å, 2 × 150 mm was employed. Column flow rate was 0.4 ml/min and elution was performed using 5 min wash time after 10μl injection and a 3 min gradient from water with 0.1% formic acid to 90% acetonitrile with 0.1% formic acid. The tandem mass spectrometry (MS/MS) transitions used for the quantitative experiments (multiple reaction monitoring, MRM) were $m/z$ 155.1–111.1 (orotic acid) and 157.1 to 113.1 (1,3-15N2 orotic acid). The acquired data were processed using the Bruker Compass Data Analysis proprietary software. Creatinine was measured using the Mouse Creatinine kit (Crystal Chem; Cat# 80350) following the manufacturer's guidelines and used to normalize orotic acid values in the urine.

Mass spectrometry analysis of argininosuccinic acid in dried bloodspots was performed as previously described (Baruteau et al, 2019b). Briefly, blood spots on a Guthrie card were dried at room temperature for 24 h. 3 mm-diameter punch from dried blood spots were used to measure argininosuccinic acid by liquid chromatography-tandem mass spectrometry (LC-MS/MS). A punch was incubated for 15 min in sonicating water bath in 100 μl of methanol containing stable isotopes used as internal standards. A 4:1 volume of methanol was added to precipitate contaminating proteins. The supernatant was collected and centrifuged at 16,000 $g$ for 5 min before separation on a Waters Alliance 2795 LC system (Waters, Midford, USA) using a XTerra® RP18, 5 μm, 3.9 × 150 mm column (Waters, Midford, USA). The mobile phases were (A) methanol and (B) 3.7% acetic acid. Detection was performed using a tandem mass spectrometer Micro Quattro instrument (Micromass UK Ltd, Cheshire, UK) using multiple reaction monitoring in positive ion mode. Data were analysed using Masslynx 4.1 software (Micromass UK Ltd, Cheshire, UK).

Mass spectrometry analysis of $^{15}N$-labelled argininosuccinic acid in liver samples was adapted from Prinsen et al (2016). Briefly, liver samples were homogenized in 400 μl of mixture of ice-cold methanol:acetonitrile:water (ratio of 5:3:2) containing internal standard (2 nmol/l, L-citrulline-d7, CDN Isotopes, Pointe-Claire, Quebec) and centrifuged at 17,000 $g$ for 20 min at 4°C. Supernatants were dried using Eppendorf® Concentrator Plus and resuspended in 0.05 M HCl. 80 μl of the resuspended mixture was mixed with 280 μl of solvent A (10 mM ammonium formate, 85% Acetonitrile + 0.15% formic acid), centrifuged at 17,000 $g$ for 5 min and filtered with 0.2 μM PTFE membrane filter (Thermo-Fisher Scientific, Rockford, IL, USA) before analysis by hydrophilic interaction liquid chromatography coupled with tandem mass spectrometry. Amino acid chromatography separation was performed in Acquity Ultra-Pure Liquid Chromatography (UPLC)-system (Waters, Manchester, UK) using Acquity UPLC BEH Amide column (2.1 × 100 mm,

1.7 μm particle size) and Van Guard™ UPLC BEH Amide pre-column (2.1 × 5 mm, 1.7 μm particle size) (Waters Limited, UK). The mobile phases were (A) 10 mM ammonium formate in 85% acetonitrile and 0.15% formic acid and (B) 15 mM ammonium formate containing 0.15% formic acid, pH 3.0. Detection was performed using a tandem mass spectrometer Xevo TQ-S (Waters, UK) using multiple reaction monitoring in positive ion mode. The dwell time was set automatically with MRM-transition of 292.2 > 116.98 and 274.2 > 70.2 for $^{15}N$-ASA and $^{15}N$-ASA anhydrides, respectively. L-Citrulline-d7 was used as internal standard control. Argininosuccinate data were analysed using Masslynx 4.2 software (Micromass UK Ltd, Cheshire, UK).

### In vivo $^{15}N$-ureagenesis in $Asl^{Neo/Neo}$ mice

Vehicle- and TB-1-treated $Asl^{Neo/Neo}$ mice received an i.p. injection of 4 mmol/kg of $^{15}N$-labelled ammonium chloride ($^{15}NH_4Cl$, Cambridge Isotope Laboratories) 20 min before terminal blood collection via cardiac puncture. Blood was collected in 1:10 volume of sodium citrate and centrifuged at 13,000 $g$ for 5 min. Plasma was collected and snap-frozen in dry ice. $^{13}C,^{15}N_2$-urea (Sigma) was added to all samples as an internal standard, and samples were derivatized in a two-stage procedure. Firstly, urea was cyclized with 1,1,3,3-Tetramethoxypropane (Sigma-Aldrich) under acidic conditions to obtain 2-hydroxypyrimidine (2HP). 2HP was then coupled with 2,3,4,5,6-Pentafluorobenzyl bromide (Sigma-Aldrich) to yield a derivative that, upon negative ion chemical ionization gas chromatography mass spectrometry, yields a negatively charged 2-HP fragment that includes the nitrogen and carbon atoms of the starting urea. Ions of mass/charge 95 (2HP from $^{12}C,^{14}N_2$-urea), 96 ($^{12}C,^{15}N,^{14}N$-urea derived from $^{15}N$-ammonium chloride or $^{13}C,^{14}N_2$-urea) and 98 ($^{13}C,^{15}N_2$-urea internal standard) were analysed, and quantified with suitable standard curves (95/98 for unlabelled urea, 96/98 for $^{15}N$-urea). Unlabelled urea concentration was calculated from the 95/98 ratio, the contribution of unlabelled urea to mass 96 calculated from the natural abundances of the atoms in the fragment ($C_4H_3N_2O^-$), and $^{15}N$-urea calculated from the 96/98 ratio minus the contribution of unlabelled urea to mass 96.

### Metabolite profiling of liver tissue by $^1H$-NMR

Livers were mechanically disrupted to extract the metabolites of interest (lipids, carbohydrates, amino acids, and other low-molecular weight metabolites) while leaving other compounds (DNA, RNA, and proteins) in the tissue pellet. Homogenization of 200 mg of frozen tissue samples was carried out in cold methanol (8 ml/g of tissue, wet weight), and cold water (1.7 ml/g of tissue, wet weight) with UltraTurrax for 2 min on ice. Four ml of chloroform per g of tissue (wet weight) were added and the homogenate was gently stirred and mixed on ice for 10 min (the solution must be monophasic). Then, additional 4 ml of chloroform per g of tissue (wet weight), and 4 ml of water per g of tissue (wet weight) were added and the final mixture was well shaken and centrifuged at 12,000 $g$ for 15 min at 4°C. This procedure separates three phases: water/ methanol on the top (aqueous phase with the polar metabolites), denatured proteins and cellular debris in the middle, and chloroform at the bottom (lipid phase with lipophilic compounds). The upper and the lower layers were transferred into glass vials, the solvents

removed under a stream of dry nitrogen, and stored at −80°C until analysis. Polar extracts were resuspended in 700 µl PBS, pH 7.4 with 10% $D_2O$ for lock procedure, and then transferred into an NMR tube. High-resolution one-dimensional (1D) spectra were recorded at 600.13 MHz on a Bruker Avance III-600 spectrometer (Bruker BioSpin GmbH, Rheinstetten, Germany) equipped with a TCI CryoProbe™ fitted with a gradient along the Z-axis, at a probe temperature of 27°C, using the excitation sculpting sequence for solvent suppression. Spectra were referenced to internal 0.1 mM sodium trimethylsilylpropionate, assumed to resonate at δ = 0.00 ppm.

### NMR data processing and statistical analysis

The spectral 0.50–9.40 ppm region of the $^1H$-NMR spectra was automatically data reduced to integrated regions (buckets) of 0.04-ppm each using the AMIX 3.6 package (Bruker Biospin GmbH, Rheinstetten, Germany). The residual water resonance region (4.72–5.10 ppm) was excluded, and the integrated region was normalized to the total spectrum area. To differentiate liver tissues through NMR spectra, we carried out a multivariate statistical data analysis using projection methods as previously reported (Soria et al, 2018).

### Enzyme activity assays

OTC enzyme activity was determined in total liver protein extracts as reported previously (Ye et al, 1996) with minor modifications. One µg of total liver protein extract (in lysis buffer: 0.5% Triton-X, 10 mM Hepes pH 7.4, 2 mM DTT) was added to 350 µl of reaction mixture (5 mM ornithine, 15 mM carbamyl phosphate and 270 mM triethanolamine, pH 7.7) and incubated at 37°C for 30 min. The reaction was then stopped by adding 125 µl of 3:1 phosphoric/sulphuric acid solution followed by 25 µl of 3% 2,3-butanedione monoxime and incubated at 95°C for 15 min in the dark. Citrulline production was determined by measuring the absorbance at 490 nm.

For ASL activity, liver samples were snap-frozen in dry ice at time of collection. Protein extraction was performed on ice. Liver samples were homogenized with the Qiagen Bead Tissue Lyser (Qiagen Manchester Ltd, Manchester, UK) in 500 µl of 50 mM phosphate buffer (pH 7.4) with EDTA-free proteinase inhibitor cocktail (Roche, Basel, Switzerland) at a frequency of 30 for 30 s. Homogenates were centrifuged at 13,000 g for 10 min at 4°C. Protein quantification of the supernatant was performed using the Pierce™ BCA protein assay kit (Thermo-Fisher Scientific, Rockford, IL, USA) according to the manufacturer's instructions. ASL activity was measured by synthesis of fumarate in an excess of argininosuccinate as previously described (Baruteau et al, 2018). Fumarate was measured with the Fumarate Assay Kit (Sigma; Cat# MAK060) according to manufacturer's instructions.

### Periodic acid Schiff staining

The staining was performed according to the Standard Operating Procedure protocol at the Histopathology laboratory, Great Ormond Street Hospital, London. Sections were dewaxed in xylene, hydrated down through graded alcohol solutions to water, incubated for 10 min in 0.5% periodic acid, rinsed in distilled water, stained for 10 min with Schiff reagent, then rinsed in distilled water. Sections

were then washed for 5 min in running tap water and counter-stained in 1% eosin for 1 min, rinsed briefly in running tap water and dehydrated through ascending grades of alcohol. Sections were then cleared in xylene and mounted. For haematoxylin and eosin (H&E) staining, liver sections were processed according to standard protocols.

For quantitative PAS staining quantification, ten random images per liver section were captured with a microscope camera (DFC420; Leica Microsystems, Milton Keynes, UK) and software (Image Analysis; Leica Microsystems). Quantitative analysis was performed with thresholding analysis using the Image-Pro Premier 9.1 software (Rockville, MD, USA).

### Hepatic glycogen determination

Liver content of glycogen was determined by using a colorimetric glycogen assay kit (Sigma-Aldrich; Cat# MAK016). Liver lysates were made by homogenization following the manufacturer's instructions using a Tissue Lyser (Qiagen). Hepatic glycogen levels were normalized for protein concentrations determined by Bradford Reagent (Bio-Rad).

### Western blotting

Liver specimens were homogenized in RIPA buffer in the presence of complete protease inhibitor cocktail (Sigma), incubated for 20 min at 4°C and centrifuged at 16,800 g for 10 min. Pellets were discarded and cell lysates were used for western blots. Total protein concentration in cellular extracts was measured using the Bradford Reagent (Bio-Rad). Protein extracts were separated by SDS–PAGE and transferred onto polyvinylidene difluoride (PVDF) membranes. Blots were blocked with TBS-Tween-20 containing 5% non-fat milk for 1 h at room temperature followed by incubation with primary antibody overnight at 4°C. The primary antibodies used were: rabbit anti-LC3B (Novus Biologicals; Cat# NB-100-2220; dilution: 1/1,000), mouse anti-p62 (Abnova; Cat# H00008878-M01; dilution: 1/1,000), mouse anti-NBR1 (Abnova; Cat# H00004077-M01; dilution: 1/1,000), rabbit anti-PYGL (Proteintech; Cat#15851-1-AP; dilution: 1/1,000), rabbit anti-NAGS (Abcam; Cat# ab65536; dilution: 1/1,000), rabbit anti-CPS1 (Abcam; Cat# ab45956; dilution: 1/1,000), rabbit anti-OTC (Novus Biologicals; Cat# NBP1-31582; dilution: 1/1,000), mouse anti-ASS1 (Abcam; Cat# ab124465; dilution: 1/1,000), rabbit anti-ASL (Abcam; Cat# ab201026; dilution: 1/1,000), rabbit anti-ARG1 (Abcam; Cat# ab91279; dilution: 1/1,000), mouse anti-β-actin (Novus Biologicals; Cat# NB600-501; dilution: 1/3,000), and mouse anti-GAPDH (Santa Cruz Biotechnology; Cat# sc-32233; dilution: 1/3,000). Proteins of interest were detected with horseradish peroxidase (HRP)-conjugated goat anti-mouse or anti-rabbit IgG antibody (GE Healthcare). Peroxidase substrate was provided by ECL Western Blotting Substrate kit (Pierce). Densitometric analyses of the western blotting bands were performed using ImageJ Software (Fiji 2).

### Electron microscopy

For EM analyses, liver specimens were fixed in 1% glutaraldehyde in 0.2 M HEPES buffer. Small blocks of liver tissues were then post-fixed in $OsO_4$ and uranyl acetate.

> ### The paper explained
>
> #### Problem
> Urea cycle disorders have high morbidity and mortality and require development of novel and more effective therapies. Ornithine transcarbamylase (OTC) and argininosuccinate lyase (ASL) deficiencies are the two most common urea cycle disorders.
>
> #### Results
> Mice carrying a Beclin-1 activating mutation have increased ammonia detoxification and treatment with the cell-penetrating autophagy-inducing Tat-Beclin-1 peptide improved phenotypic and biochemical abnormalities of mouse models of OTC and ASL deficiencies.
>
> #### Impact
> Drugs activating Beclin-1 have potential for therapy of UCD.

After dehydration through a graded series of ethanol solutions and propylene oxide tissue samples were embedded in epoxy resin and polymerized at 60°C for 72 h. From each sample, thin sections were cut with a Leica EM UC7 ultramicrotome and images acquired by FEI Tecnai − 12 (FEI, Einhoven, The Netherlands) EM equipped with Veletta CCD camera for digital image acquisition.

### Statistical analyses

Data were analysed using GraphPad Prism 5.0 software, San Diego, CA, USA. Comparisons of continuous variables between two and more experimental groups were performed using the two-tailed unpaired Student's $t$-test or one-way ANOVA with Tukey's or Dunnett's *post hoc* tests. Two-way ANOVA and Tukey's *post hoc* tests were performed to compare two groups relative to two factors. Kaplan–Meier survival curves were compared with the log-rank test. No statistical methods were used to predetermine the sample size. A minimum of $n = 5$ per group was included, and the sample size was increased if needed to achieve statistical significance. No formal randomization procedure was used but assignment of mice to treatment groups was based on mouse identification numbers, and the investigators were not blinded. Number of replicates is reported in the figure legends. Data are expressed as means ± SEM. $P$ values < 0.05 were considered statistically significant.

## Data availability

This study includes no data deposited in external repositories. All data reported in the paper are included in the manuscript or available in the Appendix.

**Expanded View** for this article is available online.

## Acknowledgements
We thank Edoardo Nusco and Carmen Lanzara (TIGEM) for technical assistance with mouse studies. This work was supported by grants of Fondazione Telethon Italy (to N.B.-P.), MIUR (PRIN2017 to N.B.-P.), NIHR Great Ormond Street Hospital Biomedical Research Centre (to J.B. and S.E.), Nutricia Metabolic Research Grant (to J.B.), London Advanced Therapy/Confidence in Collaboration award 2CiC017 (to J.B.), Medical Research Council, Grant/Award Number: MR/T008024/1 (to J.B.) and Innovate UK Biomedical Catalyst Early stage award 14720 (to J.B.). The views expressed are those of the author(s) and not necessarily those of the NHS, the NIHR or the Department of Health.

## Author contributions
LRS performed study concept and design, acquisition of data, analysis, interpretation of data and wrote the manuscript; SG performed ASL activity assay and argininosuccinate determinations; GDS performed studies in *spf-ash* mice; DPP performed studies in $Asl^{Neo/Neo}$ mice; ADA performed some Western blots; GB performed studies on $Becn1^{F121A}$ mice; EP performed EM; DP analysed $^{1}$H-NMR data; PC performed $^{1}$H-NMR; AM supervised $^{1}$H-NMR studies; YK and PBM performed $^{15}$N-argininosuccinate determinations, MO, SE and SNW participated to studies in $Asl^{Neo/Neo}$ mice; CS supervised studies on $Becn1^{F121A}$ mice; AFM supervised studies in *spf-ash* mice; JB supervised studies in $Asl^{Neo/Neo}$ mice; NB-P supervised the study, performed study concept and design, analysis and interpretation of data, and wrote the manuscript.

## Conflict of interest
The authors declare no that they have no conflict of interest.

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
