## [Review Process File · EMBO Molecular Medicine]

Beclin-1-mediated activation of autophagy improves proximal and distal urea cycle disorders.

Leandro Soria, Sonam Gurung, Giulia De Sabbata, Dany Perocheau, Angela De Angelis, Gemma Bruno, Elena Polishchuk, Debora Paris, Paola Cuomo, Andrea Motta, Michael Orford, Youssef Khalil, Simon Eaton, Philippa Mills, Simon Waddington, Carmine Settembre, Andres Muro, Julien Baruteau, and Nicola Brunetti-Pierri

DOI: 10.15252/emmm.202013158

Corresponding author: Nicola Brunetti-Pierri (brunetti@tigem.it)

Review Timeline:

Submission Date:	22nd Jul 20
Editorial Decision:	3rd Sep 20
Revision Received:	11th Nov 20
Editorial Decision:	20th Nov 20
Revision Received:	23rd Nov 20
Accepted:	25th Nov 20

Editor: Zeljko Durdevic

Transaction Report:

3rd Sep 2020

Dear Prof. Brunetti-Pierri,

Thank you for the submission of your manuscript to EMBO Molecular Medicine, and please accept my apologies for the delay in getting back to you. We have received feedback from two of the three reviewers who agreed to evaluate your manuscript. Should referee #1 provide a report, we will send it to you, with the understanding that we will not ask for an additional revision. As you will see from the reports below, both referees are positive and find the study interesting and important. However, they also raise some concerns that should be addressed in a major revision of the current manuscript. Particular attention should be given to providing additional data to strengthen the conclusions regarding the effects of TB-1 on 1) the ureagenic flux and 2) the AslNeo/Neo mice, particularly on the hepatocellular injury.

Addressing the reviewers' concerns in full will be necessary for further considering the manuscript in our journal. Acceptance of the manuscript will entail a second round of review. Please note that EMBO Molecular Medicine encourages a single round of revision only and therefore, acceptance or rejection of the manuscript will depend on the completeness of your responses included in the next, final version of the manuscript. For this reason, and to save you from any frustrations in the end, I would strongly advise against returning an incomplete revision.

We would welcome the submission of a revised version within six months for further consideration. However, we realize that the current situation is exceptional on the account of the COVID-19/SARS-CoV-2 pandemic. Please let us know if you require longer to complete the revision.

I look forward to receiving your revised manuscript.

***** Reviewer's comments *****

Referee #2 (Comments on Novelty/Model System for Author):

The manuscript by Soria et al. elaborates on earlier, elegant work by the same principal authors which revealed the importance of autophagy for the urea cycle in health and disease. The authors have now extended these studies and have made excellent use of the Becn1F121A-mouse model

in combination with two different mouse models representing inherited defects in the urea cycle, to provide convincing evidence that activation of autophagic flux is beneficial for mice. The technical quality of the experiments which includes stable isotope technology, is of the highest level in the field.

Referee #2 (Remarks for Author):

The manuscript by Soria et al. elaborates on the earlier, very original and impressive work from the same set of principal authors published in 2018 which provided the first suggestive evidence showing that stimulation of autophagic flux induces flux through the urea cycle which is especially relevant because of the occurrence of patients with genetic defects in the urea cycle. The authors have now made clever use of the *Becn1*^{F121A}-mouse model generated by Rocchi et al. in 2017, to show that constitutive activation of autophagy is associated with increased urea cycle flux in two mouse models with partial deficiencies of ornithine transcarbamylase (OTC) and argininosuccinate lyase (ASL). Overall, the experiments have been performed well but I do have a problem with the interpretation of the results especially those in the *Asl*^{NeoNeo}-mice depicted in Fig.2. My concerns are as follows:

(1.) The authors show that ureagenic flux is doubled under conditions of increased autophagic flux as induced by TB-1 (Fig.2B); (2.) the authors also show that the level of argininosuccinate measured by NMR (PS Why NMR when there are real good alternative and I would say superior methods around to measure argininosuccinate using tandem mass spectrometry for instance); and (3.) the authors show in the Supplemental Figs. that the activity of the enzyme reacting with argininosuccinate that is ASL is more or less identical under the different conditions. Taken together, the question arises: HOW CAN YOU HAVE INCREASED FLUX TO UREA AND HENCE INCREASED FLUX THROUGH THE ENZYME ASL IF THE CONCENTRATION OF THE SUBSTRATE ARGININOSUCCINATE IS LOWER!?!? One would expect to have a HIGHER concentration of argininosuccinate because that's the only way to get increased flux through ASL especially since there is no SECOND substrate involved in the ASL-reaction in contrast to the OTC-reaction in which case there is carbamoylphosphate and ornithine which together determine flux through OTC. This apparent discrepancy needs clarification. One thing that comes to mind is the question whether the concentration of ASS has been adequately determined using NMR. I would suggest to repeat this analysis using established alternative methods preferably using stable isotope technology.

Referee #3 (Remarks for Author):

Soria et al investigate the use of TB-1 peptide for the enhancement of autophagy in two mouse models of urea cycle disorders. The authors report improvement in hyperammonemia and survival in a mouse model of OTC deficiency and increased ureagenesis and improved survival in a mouse model of argininosuccinic aciduria. However, more details are needed to support these conclusions.

1) Why were only male mice used for these studies? Were similar results obtained in female mice?

2) In discussion of TB-1 therapy and *spf-ash* mice, the authors state "Taken together, these results support the therapeutic potential of activation of liver autophagy by TB-1 in combination with conventional treatments, such as ammonia scavenger drugs and L-Arg". However, in Figure 1F, the authors show ammonia levels from *spf-ash* mice treated with high protein diet after 4 days of exposure to therapy. The ammonia levels at this time point appear similar in the *spf-ash* mice

treated with arginine + benzoate and wild type mice. Moreover, there is no difference in ammonia levels between the arginine + benzoate-treated mice and the arginine + benzoate + TB-1 mice. Thus, it is unclear why the authors are concluding that TB-1 is improving hyperammonemia and has benefit beyond standard therapies? Lastly, if the spf-ash mice have similar ammonia levels as WT mice at day 4, why are the mice dying in the setting of arginine + benzoate therapy and why are they dying with the addition of TB-1? If the ammonia is improved, wouldn't long-term survival improve?

3) Similarly for Figure 1F, why did the authors only look at ammonia levels at day 4? Are there differences in ammonia levels at day 6 or 8 between WT and TB-1? The addition of this data would strengthen the argument that the therapy is improving hyperammonemia in this setting, especially if the ammonia levels were still similar to WT.

4) For the OTC studies, why was arginine rather than citrulline used for the therapy? If comparing to standard therapy, a better comparison would be between mice treated with citrulline + benzoate vs. citrulline + benzoate + TB-1. In humans with this disorder, citrulline is typically considered standard therapy.

5) The authors should provide more details in the methods section for clarification of the experimental design, especially for AslNeo/Neo mice. The authors state that all studies with AslNeo/Neo mice were done starting at day 10 and all mice were treated with a reduced protein diet and injections of benzoate and arginine. However, mice are typically not weaned until day 21. Thus, did these mice (both wild type and mutant mice) remain with the mother for the duration of the experiment? Was the reduced protein diet used to feed the dams? Were the mice fasted prior to collection of liver for glycogen assessment? If the mice were not fasted, how did the authors control for the time since last meal as this could impact hepatic glycogen stores? This is especially important given previously published data by members of this group and others suggesting that the AslNeo/Neo mice are smaller than wild type mice. Thus, how are the authors certain that the feeding behavior prior to tissue collection is similar in the WT vs. AslNeo/Neo mice especially given size differences (e.g. consumption of diet vs. breastmilk when housed with dam and time since last meal).

6) It would be helpful to the reader if the authors provided more information to support their conclusions of improved phenotype in the AslNeo/Neo mice. First, for survival, the authors state: "AslNeo/Neo mice treated only with TB-1 showed increased survival compared to vehicle-treated controls that started dying by ten days of age". However, later, they state that therapy started at day 10? Are the authors attributing the lack of deaths at day 10 to the therapy? Are they suggesting that it works this quickly? Was the death rate data at day 10 collected before or after therapy? In addition, data for survival is only shown up through day 16? How much longer do these treated mice survival beyond day 16? This data is important to support the author's conclusions of improved survival in this model.

7) Likewise, additional data about body weight and ammonia levels are needed to support the conclusions of improvements in AslNeo/Neo mice using this therapy. Specifically, the authors should provide body weight at the start of therapy and at the conclusion of therapy in all mice and ammonia levels in the various study groups. As dosing was reported to be weight-based, this data should be readily available. Likewise, it would be helpful if body weight data was also included for the spf-ash mouse study.

8) The authors report a marked reduction in hepatic glycogen. However, the data in Figure 3C

shows approximately 30% reduction? Thus, the authors may want to change the language used to describe this partial reduction in hepatic glycogen.

9) In the abstract, the authors report that the therapy "alleviated hepatocellular injury" in the AslNeo/Neo mice treated with TB-1. However, the authors do not show any data on AST, ALT or hepatomegaly in these mice. Were these phenotypes altered in mutant mice? Were they improved after therapy? If not, the authors should elaborate as to why they suspect that a mild reduction in hepatic glycogen indicates improved hepatocellular injury?

10) Figure 4 describes nuclear glycogen in the AslNeo/Neo mouse model but the authors do not comment on the significance of this finding. What is the purpose of nuclear glycogen (even in WT mice)?

11) In supplemental Figure 5, the authors present metabolomics data from the liver of AslNeo/Neo mice. Are these the only metabolites that differ among the three groups: the AslNeo/Neo with various treatments and wild type mice? The authors should include a more detailed assessment of the metabolomics data if additional metabolites differ and should provide this data in the paper. The citation of Figure 2 subunits is not right. Please correct.

Point-by-point response to reviewers**Referee #2**

*The manuscript by Soria et al. elaborates on the earlier, very original and impressive work from the same set of principal authors published in 2018 which provided the first suggestive evidence showing that stimulation of autophagic flux induces flux through the urea cycle which is especially relevant because of the occurrence of patients with genetic defects in the urea cycle. The authors have now made clever use of the *Becn1*F121A-mouse model generated by Rocchi et al. in 2017, to show that constitutive activation of autophagy is associated with increased urea cycle flux in two mouse models with partial deficiencies of ornithine transcarbamylase (OTC) and argininosuccinate lyase (ASL).*

Authors' response: We thank the reviewer for the positive feedback.

*Overall, the experiments have been performed well but I do have a problem with the interpretation of the results especially those in the *Asl* NeoNeo-mice depicted in Fig.2. My concerns are as follows:*

(1.) The authors show that ureagenic flux is doubled under conditions of increased autophagic flux as induced by TB-1 (Fig.2B);(2.) the authors also show that the level of argininosuccinate measured by NMR (PS Why NMR when there are real good alternative and I would say superior methods around to measure argininosuccinate using tandem mass spectrometry for instance);and (3.) the authors show in the Supplemental Figs. that the activity of the enzyme reacting with argininosuccinate that is ASL is more or less identical under the different conditions. Taken together, the question arises: HOW CAN YOU HAVE INCREASED FLUX TO UREA AND HENCE INCREASED FLUX THROUGH THE ENZYME ASL IF THE CONCENTRATION OF THE SUBSTRATE ARGININOSUCCINATE IS LOWER!?!? One would expect to have a HIGHER concentration of argininosuccinate because that's the only way to get increased flux through ASL especially since there is no SECOND substrate involved in the ASL-reaction in contrast to the OTC-reaction in which case there is carbamoylphosphate and ornithine which together determine flux through OTC. This apparent discrepancy needs clarification. One thing that comes to mind is the question whether the concentration of ASS has been adequately determined using NMR. I would suggest to repeat this analysis using established alternative methods preferably using stable isotope technology.

Authors' response: This is an excellent point and we are grateful to the reviewer for raising this issue. In *Asl*^{Neo/Neo} mice, TB-1 increased ureagenesis and reduced argininosuccinate despite the lack of evidence for an increased ASL activity. Although we previously found in wild-type mice that TB-1 increased ureagenesis by furnishing the urea cycle with key intermediate metabolites and preventing ammonia-induced depletion of ATP (Soria et al., 2018), the reviewer is correct that we cannot conclude whether this mechanism is also valid in ASL deficiency. We revised the discussion to reflect this important issue. Moreover, as recommended by this reviewer, we confirmed the reduction of argininosuccinate in *Asl*^{Neo/Neo} mice treated with TB-1 by GC-MS. In addition, reduction of hepatic argininosuccinate was confirmed by measurement of ¹⁵N-argininosuccinate. These new data are shown in **Fig. 2E and F** respectively, whereas the NMR-based measurements was moved to **Fig. EV4**.

Referee #3

Soria et al investigate the use of TB-1 peptide for the enhancement of autophagy in two mouse models of urea cycle disorders. The authors report improvement in hyperammonemia and survival in a mouse model of OTC deficiency and increased ureagenesis and improved survival in a mouse model of argininosuccinic aciduria. However, more details are needed to support these conclusions.

1) Why were only male mice used for these studies? Were similar results obtained in female mice?

Authors' response: Female *spf-ash* are not affected and thus, only male *spf-ash* can be used. For consistency, we next used males also for ASL-deficient mice. As reported by the Wilson's lab (Ashley et al., 2018), *Asl*^{Neo/Neo} mice have a mean survival of 22 days that is longer than the survival of male mice (shown in Fig. 3A in the paper by Ashley et al., 2018). Because our data with TB-1 have been obtained males that have a more severe phenotype, a similar or better outcome is likely to be observed in female ASL-deficient mice. Gender differences in response to TB-1 is beyond the scope for this proof-of-concept study. Nevertheless, we agree that this important issue would need to be investigated in future preclinical studies performed prior to a clinical trial.

2) In discussion of TB-1 therapy and spf-ash mice, the authors state "Taken together, these results support the therapeutic potential of activation of liver autophagy by TB-1 in combination with conventional treatments, such as ammonia scavenger drugs and L-Arg".

However, in Figure 1F, the authors show ammonia levels from spf-ash mice treated with high protein diet after 4 days of exposure to therapy. The ammonia levels at this time point appear similar in the spf-ash mice treated with arginine + benzoate and wild type mice. Moreover, there is no difference in ammonia levels between the arginine + benzoate-treated mice and the arginine + benzoate + TB-1 mice. Thus, it is unclear why the authors are concluding that TB-1 is improving hyperammonemia and has benefit beyond standard therapies? Lastly, if the spf-ash mice have similar ammonia levels as WT mice at day 4, why are the mice dying in the setting of arginine + benzoate therapy and why are they dying with the addition of TB-1? If the ammonia is improved, wouldn't long-term survival improve?

Authors' response: In our study, blood samples were collected at day 4 when all mice were still alive. It is not possible to collect blood samples as mice are getting sick. It is likely that blood ammonia levels are higher at later time points when mice start to die. Nevertheless, the reviewer is correct that ammonia levels are similar between mice treated with arginine + benzoate and mice treated with arginine + benzoate + TB-1 at the evaluated timepoint and thus, the conclusion about efficacy of TB-1 over standard therapies is based on mouse survival and not on improvement of hyperammonemia. We modified the text accordingly.

3) Similarly for Figure 1F, why did the authors only look at ammonia levels at day 4? Are there differences in ammonia levels at day 6 or 8 between WT and TB-1? The addition of this data would strengthen the argument that the therapy is improving hyperammonemia in this setting, especially if the ammonia levels were still similar to WT.

Authors' response: At day 6 or 8 most of the mice are either moribund or death, thus making additional procedures including blood drawing challenging and unethical. Therefore, it was not possible to obtain ammonia levels at these time points. As addressed in point #2, we modified the text (abstract and main text) stating that TB-1 improves survival and not hyperammonemia. Orotic acid is a biochemical hallmark of OTC deficiency and has been used in various preclinical studies with *spf-ash* mice as a clinically relevant endpoint. Therefore, we incorporated as new data in **Fig. EV2D**, the urinary orotic acid levels at baseline and after the high protein diet. We observed that Na-benzoate and L-Arg treatment did not affect urinary orotic acid increased by the high protein diet whereas TB-1 either alone or in combination with Na-benzoate and L-Arg efficiently blunted the increase in urinary orotic acid induced by the high protein diet.

4) *For the OTC studies, why was arginine rather than citrulline used for the therapy? If comparing to standard therapy, a better comparison would be between mice treated with citrulline + benzoate vs. citrulline + benzoate + TB-1. In humans with this disorder, citrulline is typically considered standard therapy.*

Authors' response: L-citrulline may be supplemented instead of L-arginine but there are no studies comparing the efficacy of the two and L-arginine is the standard treatment for OTC deficiency (please see, Häberle et al. Suggested guidelines for the diagnosis and management of urea cycle disorders: First revision. *J Inherit Metab Dis.* 2019; 42:1192-1230).

5) *The authors should provide more details in the methods section for clarification of the experimental design, especially for AslNeo/Neo mice. The authors state that all studies with AslNeo/Neo mice were done starting at day 10 and all mice were treated with a reduced protein*

diet and injections of benzoate and arginine. However, mice are typically not weaned until day 21. Thus, did these mice (both wild type and mutant mice) remain with the mother for the duration of the experiment? Was the reduced protein diet used to feed the dams? Were the mice fasted prior to collection of liver for glycogen assessment? If the mice were not fasted, how did the authors control for the time since last meal as this could impact hepatic glycogen stores? This is especially important given previously published data by members of this group and others suggesting that the AslNeo/Neo mice are smaller than wild type mice. Thus, how are the authors certain that the feeding behavior prior to tissue collection is similar in the WT vs. AslNeo/Neo mice especially given size differences (e.g. consumption of diet vs. breastmilk when housed with dam and time since last meal).

Authors' response: All these issues have been clarified in Materials and Methods. Liver glycogen accumulation appears to be independent of the type of diet (breastmilk vs. rodent diet) and it has been consistently found in mice older than 6 months (please see Baruteau et al., 2018). This increased glycogen content previously reported also by Burrage et al., 2020 is further confirmed in our study and occurs as a consequence of the reduced levels of the enzyme

glycogen phosphorylase (PYGL) that is involved in glycogen catabolism.

6) *It would be helpful to the reader if the authors provided more information to support their conclusions of improved phenotype in the AslNeo/Neo mice. First, for survival, the authors state: "AslNeo/Neo mice treated only with TB-1 showed increased survival compared to vehicle-treated controls that started dying by ten days of age". However, later, they state that therapy started at day 10? Are the authors attributing the lack of deaths at day 10 to the therapy? Are they suggesting that it works this quickly? Was the death rate data at day 10 collected before or after therapy? In addition, data for survival is only shown up through day 16? How much longer do these treated mice survival beyond day 16? This data is important to support the author's conclusions of improved survival in this model.*

Authors' response: We performed two different set of experiments in $Asl^{Neo/Neo}$ mice. First, a short-term experiment where mice received only TB-1 without any additional treatment and second, a long-term experiment in which mutant mice received a supportive treatment (to extend their survival) in combination with TB-1. For both experiments TB-1 administration started at day 10 of age and we have clarified this in the text. TB-1 is a potent and efficient activator of autophagy that quickly increases ureagenesis and ammonia detoxification capacity (Soria et al. 2018). Therefore, we expect a rapid response to TB-1. Death rate at day 10 was collected after administration of TB-1.

We have not looked at survival beyond day 16. Nevertheless, we do not believe TB-1 would provide a long-term solution for urea cycle disorder but rather it can be a treatment to prevent death occurring before more definitive treatments can be started (e.g., liver transplantation or gene therapy). For example, liver transplantation has higher risks in smaller infants whereas AAV-mediated liver-directed gene therapy has short-term efficacy if performed in young mice because of loss of vector genome due to liver growth (Baruteau et al., 2018; Ashley et al., 2018). Hence, TB-1 could be viewed as a bridge therapy allowing patients to grow until they reach an age to be safely transplanted or effectively treated with gene therapy. We have included these issues in the revised discussion.

7) *Likewise, additional data about body weight and ammonia levels are needed to support the conclusions of improvements in AslNeo/Neo mice using this therapy. Specifically, the authors should provide body weight at the start of therapy and at the conclusion of therapy in all mice and ammonia levels in the various study groups. As dosing was reported to be weight-based, this data should be readily available. Likewise, it would be helpful if body weight data was also included for the spf-ash mouse study.*

Authors' response: data on body weight and ammonia in *Asl^{Neo/Neo}* mice have been included in **Fig. EV3** and **Fig. EV5**. Data on body weight for experiments with *spf-ash* mice have also been included in **Fig. EV2**

8) *The authors report a marked reduction in hepatic glycogen. However, the data in Figure 3C shows approximately 30% reduction? Thus, the authors may want to change the language used to describe this partial reduction in hepatic glycogen.*

Authors' response: we changed 'marked' into 'partial'.

9) *In the abstract, the authors report that the therapy "alleviated hepatocellular injury" in the *AslNeo/Neo* mice treated with TB-1. However, the authors do not show any data on AST, ALT or hepatomegaly in these mice. Were these phenotypes altered in mutant mice? Were they improved after therapy? If not, the authors should elaborate as to why they suspect that a mild reduction in hepatic glycogen indicates improved hepatocellular injury?*

Authors' response: TB-1 treatment resulted in mild reductions of hepatomegaly and circulating ALT and these data are now included in **Fig EV5B** and **C**, respectively.

10) *Figure 4 describes nuclear glycogen in the *AslNeo/Neo* mouse model but the authors do not comment on the significance of this finding. What is the purpose of nuclear glycogen (even in WT mice)?*

Authors' response: this is an excellent question. The new finding of nuclear glycogen in ASA will definitely require further studies. We acknowledge this in the discussion of the revised manuscript.

11) *In supplemental Figure 5, the authors present metabolomics data from the liver of *AslNeo/Neo* mice. Are these the only metabolites that differ among the three groups: the *AslNeo/Neo* with various treatments and wild type mice? The authors should include a more detailed assessment of the metabolomics data if additional metabolites differ and should provide this data in the paper. The citation of Figure 2 subunits is not right. Please correct.*

Authors' response: To differentiate liver tissues at metabolic level through NMR spectra, we performed a multivariate statistical data analysis using projection method, as previously reported (Soria et al., 2018). Data in **Fig. S5** (now **Fig EV4**) show the metabolites with a statistically significant difference that separate classes. These metabolites were selected from the whole set of discriminating metabolites reported along the x-axis in **Appendix Fig. S1A**, using VIP value >1 and correlation loading values $p(\text{corr}) > 0.7$. Further details are reported in **Appendix Fig. S1**, with the VIP plot (**A**) and the loading plot (**B**). All the between-class discriminating metabolites are labeled on the x-axis in (**A**).

20th Nov 2020

Dear Prof. Brunetti-Pierri,

Thank you for the submission of your revised manuscript to EMBO Molecular Medicine. I am pleased to inform you that we will be able to accept your manuscript pending the following final amendments:

1) Please implement all adjustments suggested by the referee #3.

***** Reviewer's comments *****

Referee #3 (Remarks for Author):

The authors state that TB-1 leads to mild reduction in hepatomegaly in the ASL deficient mice but the figure shows no statistically significant difference in liver to body weight ratio in the treated vs. untreated mice. This statement should be clarified or removed (or the figure should be amended to indicate the difference).

The authors performed the requested changes.

Referee #3 (Remarks for Author):

The authors state that TB-1 leads to mild reduction in hepatomegaly in the ASL deficient mice but the figure shows no statistically significant difference in liver to body weight ratio in the treated vs. untreated mice. This statement should be clarified or removed (or the figure should be amended to indicate the difference).

Authors' response: As recommended, we modified the text including that the reduction of hepatomegaly showed a trend, but it is not statistically significant.

We are pleased to inform you that your manuscript is accepted for publication.

Corresponding Author Name: Nicola Brunetti-Pierri

Manuscript Number: EMM-2020-13158